# Optimization of the In Situ Biasing FIB Sample Preparation for Hafnia-Based Ferroelectric Capacitor

**DOI:** 10.3390/mi12121436

**Published:** 2021-11-24

**Authors:** Qilan Zhong, Yiwei Wang, Yan Cheng, Zhaomeng Gao, Yunzhe Zheng, Tianjiao Xin, Yonghui Zheng, Rong Huang, Hangbing Lyu

**Affiliations:** 1Key Laboratory of Polar Materials and Devices (MOE), Department of Electronics, East China Normal University, 500 Dong-chuan Road, Shanghai 200241, China; 52191213009@stu.ecnu.edu.cn (Q.Z.); 51214700096@stu.ecnu.edu.cn (Y.W.); 52181213007@stu.ecnu.edu.cn (Y.Z.); 52214700045@stu.ecnu.edu.cn (T.X.); yhzheng@phy.ecnu.edu.cn (Y.Z.); rhuang@ee.ecnu.edu.cn (R.H.); 2Key Laboratory of Microelectronics Devices and Integrated Technology, Institute of Microelectronics, Chinese Academy of Sciences, 3 Bei-tu-cheng West Road, Beijing 100029, China; gaozhaomeng@ime.ac.cn; 3University of Chinese Academy of Sciences, Beijing 100049, China

**Keywords:** hafnia-based ferroelectric, in situ biasing, FIB sample preparation technology

## Abstract

Hafnia-based ferroelectric (FE) thin films have received extensive attention in both academia and industry, benefitting from their outstanding scalability and excellent CMOS compatibility. Hafnia-based FE capacitors in particular have the potential to be used in dynamic random-access memory (DRAM) applications. Obtaining fine structure characterization at ultra-high spatial resolution is helpful for device performance optimization. Hence, sample preparation by the focused ion beam (FIB) system is an essential step, especially for in situ biasing experiments in a transmission electron microscope (TEM). In this work, we put forward three tips to improve the success rate of in situ biasing experiments: depositing a carbon protective layer to position the interface, welding the sample on the top of the Cu column of the TEM grid, and cutting the sample into a comb-like shape. By these means, in situ biasing of the FE capacitor was realized in TEM, and electric-field-induced tetragonal (t-) to monoclinic (m-) structure transitions in Hf_0.5_Zr_0.5_O_2_ FE film were observed. The improvement of FIB sample preparation technology can greatly enhance the quality of in situ biasing TEM samples, improve the success rate, and extend from capacitor sample preparation to other types.

## 1. Introduction

Since the discovery of the ferroelectric properties of hafnia-based thin film, there has been significant interest in both its fundamentals and application. Ferroelectric thin films based on HfO_2_ enable new types of computer memory and logic devices, such as ultra-low power transistors, artificial synapse tunable tunnel barriers, and non-volatile working memory [1,2,3,4,5]. However, HfO_2_-based ferroelectric thin films have a variety of crystalline phases, including monoclinic (m-, P2_1_/c), tetragonal (t-, P4_2_/nmc), cubic (c-, Fm$\bar{3}$m), and orthorhombic (o-, Pca2_1_, Pbca and Pnma) phases [6,7]. The accompanying structural phase transition is very complex. The possible transition paths from t-phase to non-centrosymmetric ferroelectric o-phase and from t-phase to m-phase have been predicted by theoretical calculation on the basis of considering surface/interface energy, stress/strain and electric field [8,9,10,11]. Understanding the structural phase transition of ferroelectric thin films under electric field, thermal field, and other external field conditions is a research hotspot [12,13].

Transmission electron microscopes (TEM), as advanced equipment, make it possible to identify different phases and structures of hafnia-based materials at the atomic scale [14,15,16,17]. In recent years, a variety of in situ techniques combined with TEM have expanded the analysis field and research content of electron microscopy with the characteristics of high spatial resolution and real-time recording and have become a new direction of TEM development [18,19,20,21,22,23]. In situ technology could provide various external fields to investigate the performance of devices by using different TEM sample holders. They are mainly divided into two kinds: in situ sample stations driven by piezoelectric ceramics and sample stations based on a micro-electro-mechanical system (MEMS) chip [18,24,25]. These two devices are inseparable from the support of focused ion beam (FIB) sample preparation technology; mature and excellent FIB sample preparation technology is the basis of successful in situ experiments.

Based on the piezoelectric-ceramics-type holder, we made some optimizations for sample preparation, making the sample more suitable for in situ biasing experiments. In this study, we proposed three tips to improve the success rate of in situ electrical samples by using FIB sample preparation technology. Using Hf_0.5_Zr_0.5_O_2_ (HZO) FE grown by high-maturity atomic layer deposition (ALD) as a model, the dynamic evolution of atomic-scale resolution under electric field conditions was successfully obtained by in situ biasing with the spherical aberration-corrected (Cs) TEM technique.

## 2. Materials and Methods

For TEM in situ observation, we made a tiny fine structure sample mounted on a Cu grid based on the traditional TEM sample preparation and FIB system (Helios G4, Thermo Fisher Scientific, Waltham, MA, USA). A TiN/HZO (5 nm)/Nb: SrTiO_3_ capacitor was prepared by ALD and then rapidly annealed at 500 °C for 30 s in N_2_ atmosphere. The P–V loop measurements were performed by a semiconductor device analyzer (WGFMU, B1500A, Agilent Technologies, Santa Clara, CA, USA). The in situ experiment was carried out on a JEM Grand ARM300F microscope (JEOL, Tokyo, Japan) operated at 300 kV with an image corrector using the PicoFemto in situ electrical holder.

## 3. Results

### 3.1. The Process of In Situ TEM Sample Preparation

Figure 1 shows the whole process of in situ TEM sample preparation using an FIB system. The sample preparation process can be summarized into five steps, which are depositing protective layers, digging pits, extracting the sample, welding the sample on Cu grid, and thinning. Figure 1a shows the full top view of the hafnia-based ferroelectric capacitors with different size of the electrodes. Figure 1b,c show the process of depositing protective layers, which can be divided into protective-layer deposition under electron beam (EB) and under ion beam (IB). The main purpose of an EB protective layer is to protect the sample from damage under IB sputtering. If necessary, a sputtering system also can be used for protection before FIB sample preparation. Figure 1d,e show the digging process, Figure 1f,g show the sample extraction process using easy lift, Figure 1h,i show the sample welding process, and finally, Figure 1j,l show the thinning process. In particular, we put forward three tips to improve the success rate of in situ experiments: 1. The first protective layer deposits carbon for identifying the interface of sample and protective layer accurately. 2. Welding the sample on the top of the column of the Cu grid; 3. Cutting the sample into comb-like shape when thinning. We will cover each tip in detail below.

### 3.2. Depositing a Carbon Protective Layer to Position the Interface

Figure 2 shows the comparison between the use of a carbon protective layer as a marker layer and another protective layer. As shown in Figure 2a and enlarged images in Figure 2b, when the first protective layer on the sample surface is carbon, it can be used as an accurate positioning basis. Because carbon has a lower atomic number, it exhibits a darker contrast under a slightly higher voltage of the scanning electron microscope (SEM). Accurate positioning can optimize the thinning effect, thus ensuring the imaging quality of the observed area and the best electrification effect in the experiment. The carbon protective layer can be taken many forms, such as a first coating, including direct deposition using a sputtering system, deposition under EB in a dual-beam system, and even direct deposition using a marker pen. If tungsten alone is used as a protective coating, the tungsten layer will show different contrast under different EB and IB condition, and then, it is difficult to distinguish the interface position of different protective layers and the sample, as shown in Figure 2c,d. Especially for capacitor samples with multilayer films, it is not clear where the sample is, and this problem also exists with the use of platinum as a protective layer.

### 3.3. Welding the Sample on the Top of the Column of the Cu Grid

Generally, the shape of copper grid is columnar and V-shaped. If it is common TEM sample preparation, the sample can be welded at any position. For the sample used in the in situ experiment of the piezoceramic driven, we propose that it can be welded at the top of the column. As shown in Figure 3, welding the sample at the top of the column has two advantages. Firstly, as shown in Figure 3a, the sample thinned with FIB can be further thinned with a Precision Ion Polishing System (PIPS) to reduce the film thickness and improve the observation quality. If the sample is welded on the side of the column, the Ar ion beam will be shielded by the grid column during PIPS thinning, while the sample welded on the top can be thinned and polished in a more comprehensive way. Secondly, Figure 3c,d are enlarged images of the regions A and B in Figure 3b, respectively. We simulated the contact position of the in situ probe on the sample. As shown in Figure 3c, during the in situ experiment, the probe range was not affected, and the tungsten probe could contact a wide range of activities. However, when the sample was welded to the side of the column, it was found that the range of activity of the tungsten probe was severely limited, and the large area on the left side of the sample could hardly contact the tungsten probe for experiment. In particular, for polycrystalline samples such as hafnium-based ferroelectric films, different regions have different crystal phases, so a large range of experimental scope is very important to ensure that regions of interest can be found. Hence, we propose to weld the sample on the top of the column, which is more conducive to the in situ electrical experiment. Even in normal transmission sample preparation, it is recommended to weld the sample to the top of the column for further polishing and optimization of the sample.

### 3.4. Cutting the Sample into “Comb” Shape When Thinning

Bad contact and discharge are two main causes of failure of in situ electrical experiments. In order to minimize the experimental failure rate, we cut the sample into the shape of a comb during the final thinning, as shown in Figure 4. The capacitor chip was divided into several small capacitors, which were divided into three small capacitor structures, as shown in the figure. Where the tungsten probe pointed to the capacitor area, only the capacitor in this area would be conductive. The schematic diagram in the figure shows that only Capacitor 2 would be conductive, and only the blue area was conductive. In this way, the risk of short circuit and leakage can be reduced as much as possible, and the success rate of in situ electrical experiment can be guaranteed.

### 3.5. In Situ Biasing Experiment of Structural Transformation in Hafnia-Based Ferroelectric Thin Film

Based on the above three tips, we successfully prepared a hafnia-based ferroelectric capacitor sample for in situ electrical experiments. The sample structure is shown in Figure 5a; the hafnia-based film was a 1:1 mixture of HfO_2_ and ZrO_2_ (Hf_0.5_Zr_0.5_O_2_, HZO) and its thickness was 5 nm. Figure 5b,c show the typical polarization versus voltage (P–V) curves and the endurance test result under successive voltage pulses with 2.5 V. The P–V loop showed a coercive voltage of 0.8 V and a remnant polarization (Pr) of about 6.5 μC/cm^2^. The hysteresis curve shape became more antiferroelectric-like, and as the field circulated, it became ferroelectric-like. This phenomenon means that there are many antiferroelectric t-phases in the pristine state of the thin film. With the electric field cycling, the antiferroelectric phases changed into ferroelectric o-phases, which is similar to previous reports [26,27,28]. As the electric field circulated further, the polarization decreased to about 4 μC/cm^2^ after 10^8^ cycles, which is a characteristic of fatigue. It has been suggested that fatigue may be due to transition to a non-ferroelectric m-phase [29]. After understanding the macroscopic electrical properties of the sample, we further conducted an in situ electrical experiment on the sample, and the schematic diagram of the in situ experiment is shown in Figure 5d. A diagram of the upper left corner shows the schematic of the DC voltage–time profile applied on the probe.

Figure 6a shows the high magnification capacitor structure with a t-grain between the two electrodes projected along the T[110] direction. This grain was about 36 nm long and slightly trapezoidal. Figure 6b contains enlarged high-angle annular dark-field (HAADF) and annular bright-field (ABF) images of the yellow boxes in Figure 6a. In this HAADF image, only heavy atoms such as Hf and Zr can be identified. In addition, the light element sensitive atomic resolution scanning transmission electron microscopy- annular bright-field (STEM-ABF) technique was used to detect oxygen atom information. The corresponding ABF experimental image of this grain shows oxygen in the middle of the heavy atoms. Figure 6c is the schematic of the HZO unit cell of the t-phase projected along the T[110] zone axes. The orange and red solid balls correspond to the Hf/Zr and O atoms. The experimental and simulated results were completely consistent, as shown in the inset of the simulated result. The quality of this capacitor slice made by the above method is good and can be used in in situ electrical experiments. After the electric field cycle, the change of electrification process was very rapid, and there was a certain jitter in the electrification process. At the end of the cycle, the t-phase grain was found to have completely transformed into a m-phase grain. Figure 6d shows the grain after electric field cycling, which was identified as an m-phase grain along the M[100] direction. The shape and size of the grain hardly changed, and the phase structure of the grain changed as a whole. Figure 6e contains enlarged HAADF and ABF images of the yellow boxes in Figure 6d, and Figure 6f is the schematic. The illustrations in the upper right corner of Figure 6b,e are the electron diffraction patterns obtained by using fast Fourier transformation (FFT). Structural change can be clearly observed from FFT change. The experimental results agree well with the simulation. The phase transformation process before and after the in situ electric field is consistent with the previous macroscopic properties, which proves that one of the causes of fatigue is the transformation of part of the antiferroelectric and ferroelectric phases to the non-ferroelectric phase. There has been much speculation in previous reports about the fatigue of hafnium-based ferroelectrics, the most recognized being the cause of the transition to the m-phase of non-ferroelectrics [29]. The transition from t-phase to o-phase has been verified theoretically and experimentally [30,31]. The direct current used in this experiment could cause a direct failure. In the actual sample, the process from awakening to fatigue appears, which is regarded as a transition from t to o to m phase.

Figure 7 shows another case which the m-phase grain did not change under the action of electric field. Figure 7a shows the grain shape of m-phase before and after electrification. Figure 7b shows the atomic model in m-phase M[001] orientation, corresponding to the enlarged HAADF and ABF images in Figure 7c. It can be shown that m-phase was stable under the action of electric field, and m-phase transformation would not occur during the electric field cycle.

## 4. Discussion and Conclusions

In this paper, we propose three methods to improve in situ sample preparation. Each method provides some improvement in sample preparation in three aspects: thin enough for better observation, more stable and more convenient for probe contact, and less leakage after contact. For the first point, the carbon layer positioning can help us to thin the sample to the upper electrode precisely, which is very effective for thin multilayer films, especially polycrystalline films. The sample prepared by FIB is wedge-shaped, with the thinnest at the front. With the positioning of the carbon protective layer, all the observed film areas can be thin, and the thickness of the sample will greatly affect the quality of observation. For the second point, welding the sample on the top of the B column provides the full sample range of the probe, so that the probe is not affected by the Cu grid. There are about 200 grains in the sample range of about 10 microns. The polycrystalline orientation is random, and the contact range of almost all grains can greatly improve the success rate. In addition, both sides of the bottom of the sample are welded to the Cu grid, providing a more stable support for the sample and reducing the deformation caused when the probe contacts the sample. Finally, cutting the sample into a comb shape is used to reduce leakage. When the capacitor is segmented, local leakage and other problems will not affect the whole sample. If the experiment of a single cut capacitor fails, we can still continue the experiment on the other cut capacitors, which further improves the success rate of a single experiment. In summary, we propound three tips to improve the success rate of in situ experiments. After the implementation of the above three tips, we successfully prepared HZO film samples that could be used for in situ experiments. The obtained knowledge would be beneficial for optimizing the in situ biasing TEM sample preparation method, and therefore may be applicable to (and hence can be considered in) other types of device samples, with much broader implications.

## Figures and Tables

**Figure 1 micromachines-12-01436-f001:**
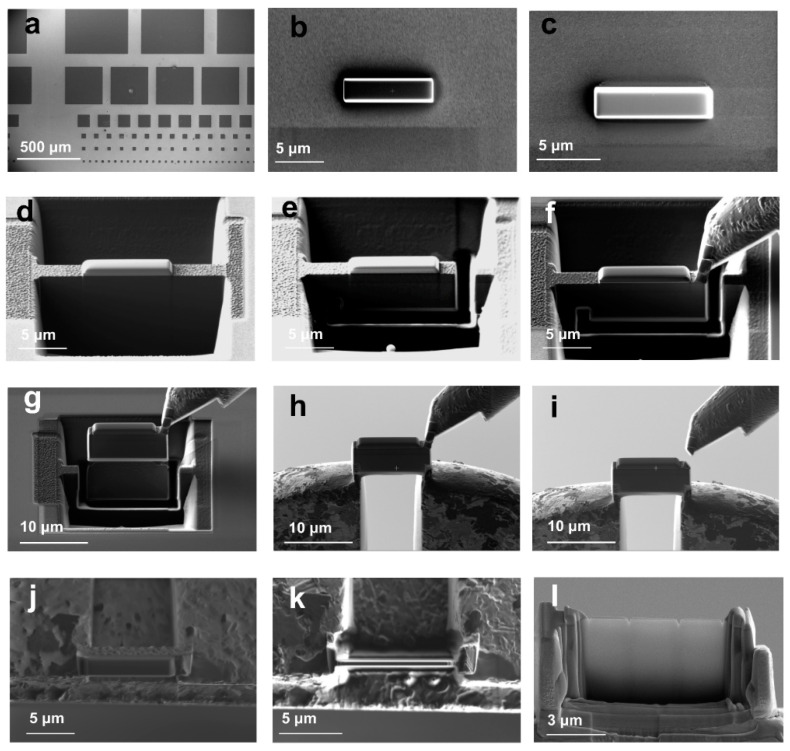
Focused ion beam (FIB) sample preparation step diagram. (**a**) Full top view of the capacitors with different size of the electrodes. (**b**) Deposited carbon as protective layer. (**c**) Deposited tungsten as protective layer. (**d**) Dig holes on both sides. (**e**) U-cut. (**f**) Weld easy lift to capacitor slice. (**g**) Lift out capacitor slice. (**h**) Weld capacitor slice on the Cu grid. (**i**) Cut the connection. (**j**) Cut under high current condition. (**k**) Cut under low current condition. (**l**) Cut into three comb-like capacitor slices.

**Figure 2 micromachines-12-01436-f002:**
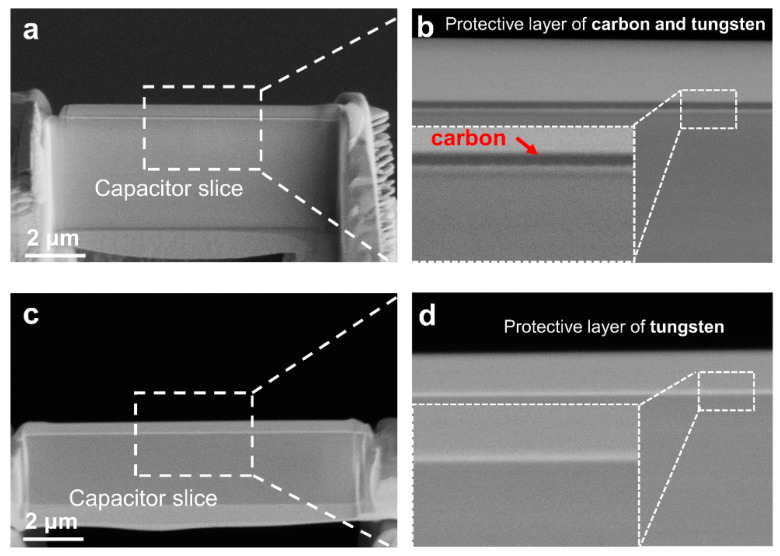
Comparison of different protective layers in thinning. (**a**) The sample is first covered with a protective carbon layer. (**b**) Magnified partial sample of a. (**c**) The sample is covered with the protective tungsten layers. (**d**) Magnified partial sample of (**c**).

**Figure 3 micromachines-12-01436-f003:**
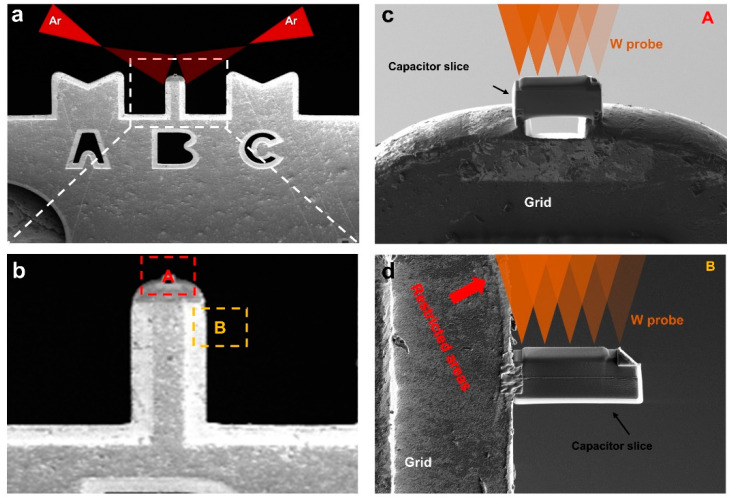
Advantages of welding the sample to the top of the column. (**a**) Simulation of polishing sample in Precision Ion Polishing System (PIPS). (**b**) Magnified column section of Cu grid. (**c**) The case that sample welded at the top of the column. (**d**) The case that sample welded at the side of the column.

**Figure 4 micromachines-12-01436-f004:**
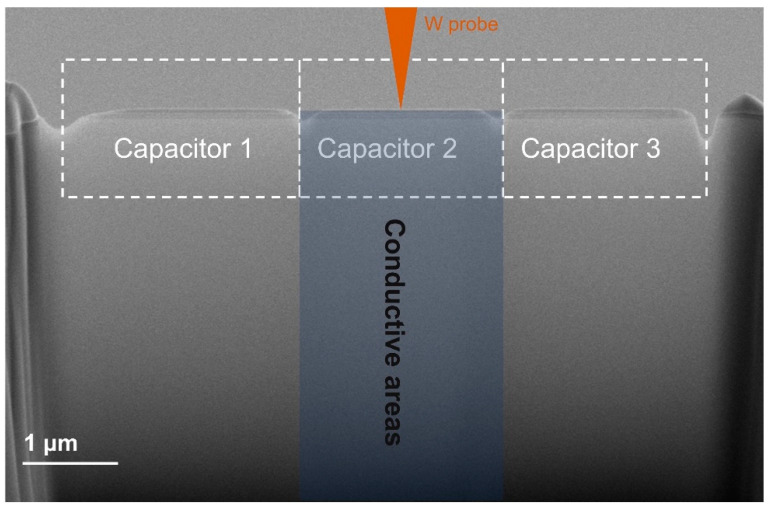
Final view of the comb-like capacitors for in situ electrical and structural characterization in transmission electron microscopy (TEM).

**Figure 5 micromachines-12-01436-f005:**
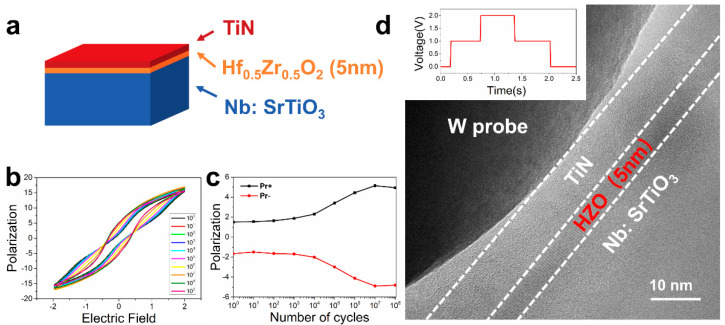
The Hf_0.5_Zr_0.5_O_2_ (HZO) thin film for in situ electrical experiments. (**a**) Schematic diagram of the HZO-based ferroelectric thin film capacitance structure for experiment. (**b**) Typical P–V curves of HZO(5nm) capacitor. (**c**) The tendency of Pr to cycles under 2.5 V. (**d**) High resolution TEM (HRTEM) image shows the in situ biasing process for a local area in the HZO thin film with the aid of tungsten probe.

**Figure 6 micromachines-12-01436-f006:**
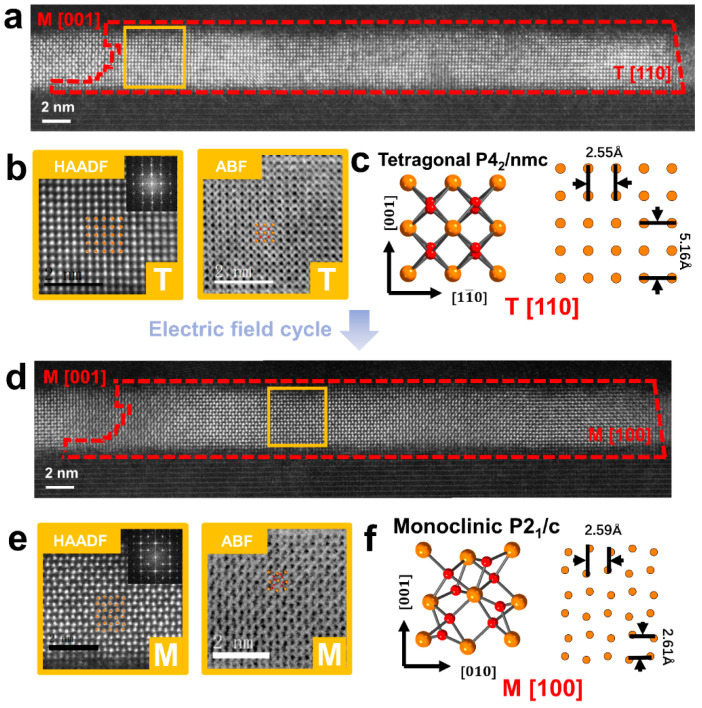
Atomic structure changes in phase transition region before and after electrification. (**a**) High magnification capacitor structure before electrification. (**b**) Enlarged high-angle annular dark-field (HAADF) and annular bright-field (ABF) diagrams of t-phases. (**c**) Atomic model of tetragonal phase. (**d**) High magnification capacitor structure after electrification. (**e**) Enlarged HAADF and ABF diagrams of m-phases monoclinic phase. (**f**) Atomic model of monoclinic phase.

**Figure 7 micromachines-12-01436-f007:**
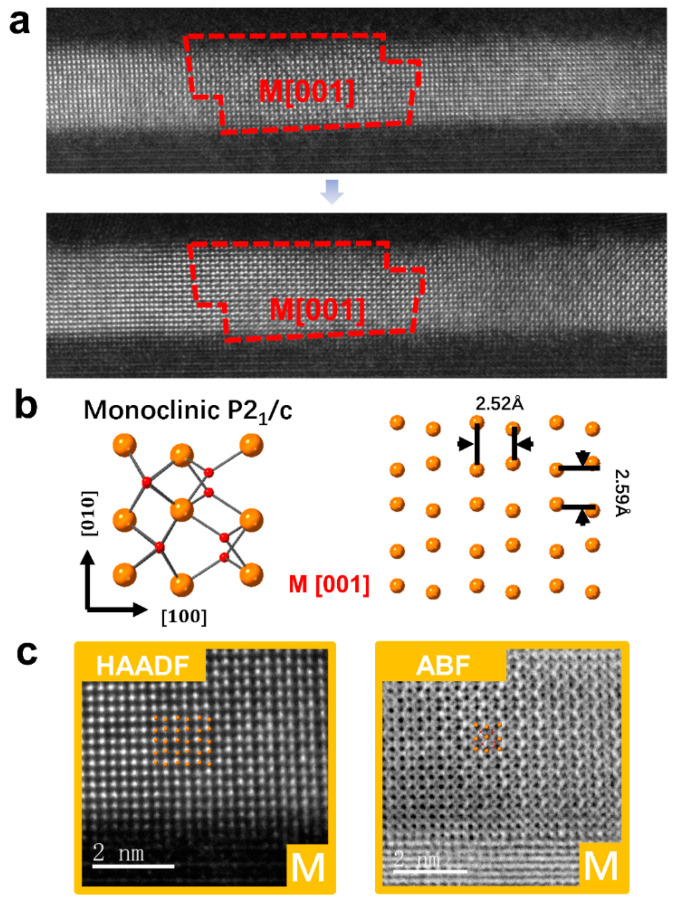
Regions where the phase structure did not change during electrification. (**a**) Low magnification image of grain that did not undergo phase transformation. (**b**) Atomic model of monoclinic phase. (**c**) Enlarged HAADF and ABF diagrams of m-phases.

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
