# Peer review of "Optimization of the In Situ Biasing FIB Sample Preparation for Hafnia-Based Ferroelectric Capacitor"

_micromachines, 2021, doi:10.3390/mi12121436_

Round 1

Reviewer 1 Report

You describe very clear the background and the methods as well as close link to electrical characterization. Also the results are very good and clear confirming other cited references.

But you don't elaborate deeper how much you improve sample preparation. If possible, please compare it with a reference. What exactly is the outcome of the improvement (enhanced resolution, better measurement success rate,....)? If it is novel and no one has done before then please point out that this analysis has been done for the first time.
What remains open for me is whether the preparation method or the result (e.g. TEM analysis of in-situ fatique) is in focus or maybe also both. Consider to adjust the title to the message.   

Reviewer 2 Report

The article is dedicated to a very interesting and up to date theme. The authors describe interesting approaches in the field of FIB sample preparation technology. However, the text itself contains some weak points, which should be corrected before publishing.

- First of all, the main aim and the main results of the investigation are unclear. According to the authors: "we made some optimizations for sample preparation, making the sample more suitable for in-situ biasing experiments". What does it mean? Are the optimizations for sample preparation proposed by authors critically important for successful in-situ biasing experiments? Is the sample not suitable for in-situ biasing experiments without author's preparation approaches? Is it possible to provide the measurement without these sample preparation stages? The difference between measurements with and without author's steps should be pointed out, and discussion of this difference should be provided in the "discussion".

- There is no discussion in the text, only the conclusions.

- The HZO films investigated by authors are polycrystalline. From the other side, HZO films on SrTiO3 can be prepared in a monocrystal form, especially by the ALD, and monocrystal films are more desirable for applications. What is the reason to investigate the polycrystalline films?

- Fig.5 is not suitable for publication - graphs on insets are too small and invisible.

- In fig.6 captions "e" and "f" should be replaced.

Reviewer 3 Report

In their work, Zhong and co-authors explain an optimized method for the preparation of Hafnia based ferroelectrics for TEM measurements using a FIB method. Their work is sound, and while the new science is relatively limited, these types of publications are very useful to the broader community as it allows a great deal of time to be saved by having a detailed methodology which can be followed for good sample preparation, and I do believe it is worthwhile to publish.

I do have a few small concerns as follows:

a.) In Fig. 1, the numbers (i.e. magnification, scale bar, etc) are almost impossible to read. It may be better to change these images to stylistically match Figure 2, etc, so that the scale of the images can be clearly read. 

b.) In figure 6b, it would be very useful to show the FFT (or diffraction pattern) as is done in Fig. 6e, as these are directly comparable images, and this would more clearly show the structural change between the before/after electrification. 

Round 2

Reviewer 2 Report

The authors have made constructive additions to the text, and have answered all questions. The article can be published in present form.